# Sirtuin 1: Endocan and Sestrin 2 in Different Biological Samples in Patients with Asthma. Does Severity Make the Difference?

**DOI:** 10.3390/jcm9020473

**Published:** 2020-02-09

**Authors:** Zoe Tsilogianni, Jonathan R Baker, Anastasia Papaporfyriou, Andrianna I Papaioannou, Evgenia Papathanasiou, Nikolaos G Koulouris, Leah Daly, Kazuhiro Ito, Georgios Hillas, Spyridon Papiris, Petros Bakakos, Stelios Loukides

**Affiliations:** 12nd Department of Respiratory Medicine, Medical School of National and Kapodistrian University of Athens, “Attikon” Hospital, 12462 Athens, Greece; zoetgr@yahoo.gr (Z.T.); papaioannouandriana@gmail.com (A.I.P.); evgjenaki11@yahoo.com (E.P.); papiris@otenet.gr (S.P.); loukstel@med.uoa.gr (S.L.); 2Airway Disease, National Heart and Lung Institute, Imperial College, London SW3 6LY, UK; jonathan.baker@imperial.ac.uk (J.R.B.); kaz@pulmocide.com (K.I.); 31st Department of Respiratory Medicine, Medical School of National and Kapodistrian University of Athens, “Sotiria” Hospital of Chest Diseases, 11527 Athens, Greece; dranastp@gmail.com (A.P.); koulnik@med.uoa.gr (N.G.K.); 4Pulmocide Ltd., London SW7 2PG, UK; Leah@pulmocide.com; 55th Respiratory Medicine Department, Sotiria Hospital, 11527 Athens, Greece; ghillas70@yahoo.gr

**Keywords:** Asthma, inflammation, severity, Endocan, Sestrin 2, Sirtuin 1

## Abstract

Background: Sestrin 2, Endocan, and Sirtuin 1 are distinct molecules with some biologic actions associated with asthma pathophysiology. The aim of the present study was to determine the molecular level differences attributable to underlying asthma severity. Methods: We initially recruited 85 asthmatics with a wide spectrum of severity. All of the patients were optimally treated according to current guidelines. Demographics, test results of lung function, and treatment regimes of all patients were recorded. Sestrin 2, Endocan, and Sirtuin 1 were measured in different biological samples (sputum with two processing methods and serum). Results: A total of 60 patients (35 with severe asthma) were analyzed, since 25 patients failed to produce an adequate sample of sputum. Patients with severe asthma showed significantly higher values for Sestrin 2 [pg/mL], measured in both sputum supernatant and cell pellet, compared to those with mild to moderate asthma [9524 (5696, 12,373) vs. 7476 (4265, 9273) *p* = 0.029, and 23,748 (15,280, 32,742) vs. 10,084 (3349, 21,784), *p* = 0.008, respectively]. No other significant differences were observed. No significant associations were observed between biomarkers, inflammatory cells, and lung function. Conclusion: Sestrin 2 is increased in patients with severe asthma as part of a mechanism that may modify structural alterations through the imbalance between oxidative stress and antioxidant activity.

## 1. Introduction

Inflammation and structural and functional abnormalities within the airways are key features of asthma [1]. Distinct molecular mechanisms and biomarkers in different biological samples have been identified and linked to clinical asthma phenotypes [2]. Although these processes are well-documented, their expressions vary across the heterogeneous spectrum of asthma severity. Severe refractory asthma (SRA) is a heterogeneous disorder characterized by persistent symptoms despite being treated with high-dose inhaled steroids (ICS) [3]. The poor response to steroids might be partly attributed to tissue repair, remodeling, and airway inflammation, which are amplified in SRA [4]. The classical approach of T2 high and T2 low asthma remains controversial, since some other molecules that are not traditionally attributed to the aforementioned dichotomous process may also be involved in the disease pathogenesis, and may also represent future treatment targets [5].

Silent information regulator 1, also called Sirtuin 1 (SIRT1), is a class-III histone deacetylase that exerts both anti-inflammatory and antiaging effects. A number of studies have reported that the activity and expression level of SIRT1 are decreased in patients with asthma [6,7]. Furthermore, several studies used ovalbumin-induced asthmatic mice models to indicate that activators of SIRT 1 may lower the inflammatory process [8]. However, the mechanisms underlying decreased SIRT1 expression and function are still unclear.

Sestrins (Sesns) have been identified as a family of highly conserved, stress-inducible proteins that are strongly upregulated by various stresses, including DNA damage, oxidative stress, and hypoxia [9]. Sestrin 2 belongs to the family of Sesns (which constitute three proteins, Sesns 1–3) and has antioxidant activities, such as restoring the peroxidase activity of overoxidized peroxiredoxins and activating nuclear factor erythroid 2-related factor, which is a potent antioxidant gene inducer [10]. Currently, data concerning asthma and Sesns are limited.

Endothelial cell-specific molecule 1, also called Endocan, is a dermatan sulfate proteoglycan, which is expressed by endothelial cells in the alveolar walls of the lung and kidneys [11]. High Endocan levels are associated with endothelial dysfunction and inflammation. Its role in the pathogenesis of asthma remains limited since it has been studied only in children where increased serum levels were connected to either the underlying severity or poor lung function [12].

Considering that severe asthma is a heterogeneous and complex disease with variable response to standard treatment, while precision medicine advocates the identification of treatable traits, linking them to therapeutic approaches that target specific inflammatory mechanisms would be of great importance in identifying whether the aforementioned inflammatory molecules are implicated in the severity of the disease. Considering all the aforementioned points, we attempted to determine whether the aforementioned biomarkers differed in severe asthma compared to mild to moderate asthma using different biological samples, and to elucidate whether any of the biomarkers are associated with either inflammatory cells or lung function.

## 2. Methods

### 2.1. Subjects

A total of 60 patients suffering with asthma who were followed up in the asthma clinics of two tertiary hospitals (1^st^ Respiratory Medicine Department, University of Athens, Sotiria Hospital, Athens, Greece, and 2^nd^ Respiratory Medicine Department, University of Athens, “Attikon” Hospital, Chaidari, Greece) were included in the study. All patients were optimally treated for at least 6 months. Severe asthma was diagnosed using the ERS/ATS guidelines. However, the selection of the dose of ICS (low, medium, and high) was based on the Global Initiative for Asthma guidelines [13]. Patients with severe asthma received high doses of ICS, compared to those with mild to moderate asthma. Compliance was assessed through the electronic prescription system. Patients with chronic obstructive pulmonary disease (COPD) or any other respiratory disease, as well as those with malignancy, heart, renal, liver, or collagen disease, were excluded. Patients with a respiratory tract infection or an asthma exacerbation in the past 8 weeks were also excluded.

After the patients agreed to participate in the study, their demographic and functional characteristics, including smoking status, atopy, body mass index, and current medication, were recorded. All patients signed an inform consent [Ethics approval: Attiko Hospital: Number 426/4/2013].

### 2.2. Sputum Induction

Sputum was induced using all the modifications for safe measurements according to the underlying asthma severity, as previously described in [14,15]. Briefly, patients inhaled 3% saline at room temperature nebulized by an ultrasonic nebulizer (De Vilbiss Co, Heston, UK) at the maximal saline output (4 mL/min) for 15 min. Patients were encouraged to expectorate at a 3-min interval. Sputum samples were collected within 2 h in order to obtain selected plugs. At least 70 mg of sputum plug was necessary in order to consider the specimen suitable and proceed to the next steps. Sputum samples, after the addition of protease and phosphatase inhibitors, were homogenized and mixed vigorously using a vortex mixer to disperse the cells in 0.01% dithiothreitol to the amount four times the sputum weight. Afterward, the same amount of phosphate-buffered saline (PBS) was added, followed by centrifugation at 790 g at 4 °C for 10 min to obtain a supernatant. After obtaining the supernatant, 1 mL PBS was added to the cell pellet, from which 50 μL sample was taken and the same amount of Trypan Blue stain was added for the measurement of differential and total cell count by a hemocytometer (Neubauer chamber). Measurement was performed by two observers, both blind to subjects’ clinical characteristics. A specimen was considered unsuitable if the recorded squamous cells were more than 10% of the total cell count. Microscope slides in Cytospin (Shandon, Runcorn, UK) were prepared by PBS dilution for the total cell count measurement. Afterward, the specimen underwent similar centrifugation in order to collect the cell pellet. All specimens were stored at −80 °C for mediator assessment. The microscope slides, after drying for two hours, underwent May-Grunwald and Giemsa staining for cell determination and measurement. Biomarkers were measured in both sputum supernatant and cell pellet.

### 2.3. Blood Sampling

Blood samples were collected in BD Vacutainer Plus Plastic Serum and SST Tubes, which are coated with silicone and micronized silica particles to accelerate clotting. Samples were then centrifuged at 2500 g for 15 min at room temperature, and serum supernatants were aliquoted and immediately stored at −80 °C until measurement.

### 2.4. Lung Function

Forced Expiratory Volume in 1^st^ sec (FEV_1_) and Forced Vital Capacity (FVC) were measured using Master Screen Body (Viasys Healthcare, Jaeger, Hochberg, Germany) according to the American Thoracic Society guidelines [16]. All reported values were measured post bronchodilation.

### 2.5. Atopic Status

A positive skin prick test (mean wheal diameter of 3 mm or greater) to any of the 20 common aeroallergens, in mites, grass, trees, fungus, and domestic animals, guided by clinical symptoms was used to confirm atopy.

### 2.6. Mediator Assays

SIRT1 protein was detected by Western blotting using a rabbit-derived, anti-SIRT1 antibody (No. 5322) (Sigma-Aldrich, Saint Louis MO, USA). Proteins were extracted from sputum cell pellets using 50 μL of RIPA buffer (150 mM NaCl, 1.0% IGEPAL CA-630, 0.5% sodium deoxycholate, 0.1% SDS, and 50 mM Tris, pH 8.0; MilliporeSigma, Hertfordshire, UK) added with protease inhibitor cocktail (Roche, Welwyn Garden City, UK). The SIRT1 at 120kDa in serum was determined as previously reported. Samples were analyzed using SDS-PAGE (Thermo Fisher Scientific) and detected with Western blot analysis by chemiluminescence (ECL Plus; GE Healthcare, Hatfield, UK) [17].

Sestrin 2 and Endocan were measured using Elisa kits (Aviva Systems Biology Corp., San Diego, CA, USA. and Abcam PLC, Cambridge, UK, respectively). For Sestrin 2, all samples (serum, sputum supernatant, and cell extracts) were diluted in the ratio of 1:2 with diluent. For Endocan, serum was diluted in the ratio of 1:2 and sputum supernatant was diluted in the ratio of 1:1 with diluent. Detection ranges were 0.156–10 ng/mL and 31.2–2000 pg/mL. Samples below range were given the value for the next value from the bottom of standard on standard curve (0.078 ng/mL and 15.6 pg/mL respectively) for statistical analysis.

## 3. Statistical Analysis

Categorical variables are presented as n (%), whereas numerical variables are presented as mean ± standard deviation (SD) or median (interquartile ranges) for normally distributed and skewed data, respectively. The normality of distribution was checked with the Kolmogorov-Smirnov test. Differences in numerical variables between two groups were evaluated with unpaired *t*-tests or Mann-Whitney U-tests for normally and skewed data, respectively, whereas comparisons of proportions were performed using chi-square tests. Simple correlations were performed with Spearman’s correlation coefficient since the data were not normally distributed. Statistical analysis was performed using SPSS 17.0 for Windows (SPSS Inc., Chicago, IL, USA), and graphs were created using GraphPad Prism 6 (GraphPad Software, Inc., La Jolla, CA, USA). *p*-values <0.05 were considered statistically significant.

## 4. Results

Demographic characteristics of study participants are provided in Table 1.

From the initially recruited subjects, 25 failed to produce an adequate sample of sputum and were excluded from the final analysis. Severe asthmatics significantly differed in terms of lung function impairment as assessed by FEV_1_ % pred, and also in the percentage of eosinophils and neutrophils in induced sputum.

Sestrin 2 and Endocan levels in different biological samples are presented in Table 2.

Sestrin 2 levels significantly differed in favor of severe asthma in both sputum supernatant and cell pellet (Figure 1A,B). No significant differences were observed for Sestrin 2 serum values and Endocan in either sputum or serum. Regarding the use of maintenance oral Cs or the use of biologics, still no significant differences were observed. No significant correlations were detected for both biomarkers with the inflammatory cells and the lung function indices.

Cell pellet SIRT1 mostly degraded, and SIRT1 at 120kDa was not detectable. Serum SIRT1 levels (expressed as optical density in Western blot) did not differ between patients with mild to moderate asthma and those with severe asthma (0.032 ± 0.02 vs. 0.047 ± 0.03, *p* = 0.414, Figure 2). Similarly, no significant difference was observed between patients who did and those who did not receive oral corticosteroids or biologicals. No significant correlations were detected between serum SIRT1 levels and lung function (FEV_1_% pred, r_s_ = −0.071 *p* = 0.596, FEV_1_/FVC ratio, r_s_= −0.169, *p* = 0.217) or between SIRT1 and the inflammatory cells in induced sputum (eosinophils%, r_s_ = 0.019, *p* = 0.884, and neutrophils %, r_s_ = −0.098, *p* = 0.456).

## 5. Discussion

In this study, we demonstrated that Sestrin 2 levels were higher in both sputum supernatant and cell pellet of patients with severe asthma compared to those with milder forms of the disease, whereas neither Endocan nor SIRT1 serum levels differed between patients with different asthma severity. No significant associations were observed between the aforementioned biomarkers and indices of lung function, inflammatory cells, and treatment regimes.

In this study, we used different methods to process biological samples. Considering sputum analysis, we used the traditional method of sputum supernatant as well as the cell pellet method. The latest method allowed us to analyze and detect concentrations of the study biomarkers at a different level: the cellular one. We failed to detect SIRT1 in both cell pellets and supernatants, while Sestrin 2 and Endocan were detectable in all samples. The aforementioned differences may be attributed either to technical assessments or to some processing issues, which may have altered the results in different ways. At the same time, they provided us with a multiassessment-based process in order to better understand the biological behavior of various biomarkers at different cellular and airway levels.

Endocan is a novel proteoglycan expressed by pulmonary endothelial cells. It is involved in a multiprocess which is characterized by different degrees of endothelial damage and microvascular inflammation, and through the activation of adhesion molecules, it mediates leukocyte trafficking and adherence to the endothelium. Considering all the aforementioned points, we can state that in diseases such as asthma, Endocan could reflect part of the aforementioned pathophysiological processes, since it is known to be involved in the disease pathogenesis. [18]. Currently, limited data exist with regard to the role of Endocan in clinical settings of asthma. A study that recruited asthmatic children with a broad spectrum of severity showed increased concentrations of Endocan in the serum of asthmatic children compared to healthy ones. In the same study, a negative association between Endocan serum levels and lung function impairment was observed. In our study, we aimed to determine whether Endocan was upregulated in different biological samples in severe asthma compared to milder forms of the disease. Our results did not meet our expectations since no significant differences were observed for both systemic and airway assessment. However, if we consider angiogenesis to be a complex multiphase process, potentially involving a great number of mediators, it is quite difficult to define the specific role or the contribution of each mediator. Furthermore, there are two limitations to the study. The first limitation is the site of the selected assessment. If the vascular remodeling process is the consequence then biopsy is the preferred sample. The second limitation is based on the possible predefined association with vascular endothelial growth factor, which is clearly the most potent angiogenetic factor in the asthmatic vascular process [19,20].

SIRT1 has a regulatory role in the airways [21]. When we initially designed the current study, our main focus was the measurement of SIRT1, since we expected to have either increased or decreased levels in severe asthma. Based on our initial hypothesis, we speculated that the aforementioned alterations in SIRT1 levels would clearly discriminate severe asthma from the milder forms of the disease. Furthermore, the aforementioned differences would be detectable in both airways and serum. Interestingly, both hypotheses were not confirmed by the results. SIRT1 was not detectable in sputum; this may be attributed either to the biological process of SIRT1 or to the sampling process. Assessing SIRT1 in serum by optical density failed to show any differences in favor of severe asthma. Published data support that defective SIRT1 increased airway inflammation [7], while the activation of SIRT1 by different processes had the opposite result [8]. In a clinical study, protein expression of SIRT1 activity obtained in peripheral blood cells from patients with severe asthma was decreased. This decrease was associated with the degree of airflow limitation, as well as with some biomarkers expressing the upregulation of T2 responses [6]. The aforementioned alterations were not observed in our study. We believe that the main aspects that drive our results are attributed either to the optimal treatment of asthmatics or to the preserved control of the disease, and they may be further justified by the absence of intense airway inflammation as assessed by sputum cells. Regarding serum assessment, this is quite different from what we have previously observed in COPD [17], since the serum is not the ideal tool for obtaining inflammatory differences in asthma. In contrast to the aforementioned speculation, some other data, mainly obtained from animal studies, support that increased levels of SIRT1 promote inflammation that mainly characterized the T2 process [22,23]. Whether these results are more compatible with our data remains controversial, since our study lacks information regarding T2-related mediators. Extending the speculative approach, we can initiate a hypothesis that the absence of significant differences in optimal density may just represent increased protective levels in severe asthma.

The only positive finding from the current study was the increased levels of Sestrin 2 in both sputum supernatant and cell pellet in patients with severe asthma. The aforementioned findings indicate a possible implication of Sestrin 2 in either the airway or in the cellular part of inflammation. To the best of the authors’ knowledge, this is the first study aimed at determining the role of Sestrin 2 in asthma severity. A recently published study showed increased levels of plasma Sestrin 2 in asthmatics during exacerbation and in recovery compared to healthy subjects [24]. In the same study, an independent association between Sestrin 2 and FEV_1_ % pred was observed. Sestrin 2 seems to protect against oxidative stress since its inactivation may lead to both accumulation of reactive oxygen species and upregulation of oxidative stress. In an animal model, a similar inactivation-based process upregulated some indices, which were involved in airway remodeling, such as transforming growth factor β [25]. Interestingly, in a similar animal study, the mutational inactivation of Sestrin 2 prevented the development of cigarette-smoke-induced pulmonary emphysema by upregulating platelet-derived growth factor receptor β PDGFRβ expression [26]. Taking into consideration the aforementioned data and combining them with our results, we believe that Sestrin 2 protects against oxidative stress and some of the features of airway remodeling. This is partially supported by the absence of any association between Sestrin 2 and inflammatory cells obtained by induced sputum. Further studies are needed in order to elucidate any associations of Sestrin 2 with more direct markers of oxidative stress. Similarly, nonsignificant associations were observed between the biomarkers and the indices of lung function tests and treatment regimes. This may be attributed to the well-controlled status of asthma in the optimally treated patients and the suppression of the cellular-derived airways inflammatory process.

In conclusion, the current study failed to recognize a role for both Endocan and Sirtuin 1 in terms of asthma severity. The positive finding refers to Sestrin 2, which was increased in both sputum supernatant and cell pellet in patients with severe asthma. Considering the aforementioned points, we may speculate that Sestrin 2 is possibly part of a mechanism that may modify structural alterations through the imbalance between oxidative stress and antioxidant activity. However, in order to strengthen the aforementioned speculation, an experimental study must be conducted.

## Figures and Tables

**Figure 1 jcm-09-00473-f001:**
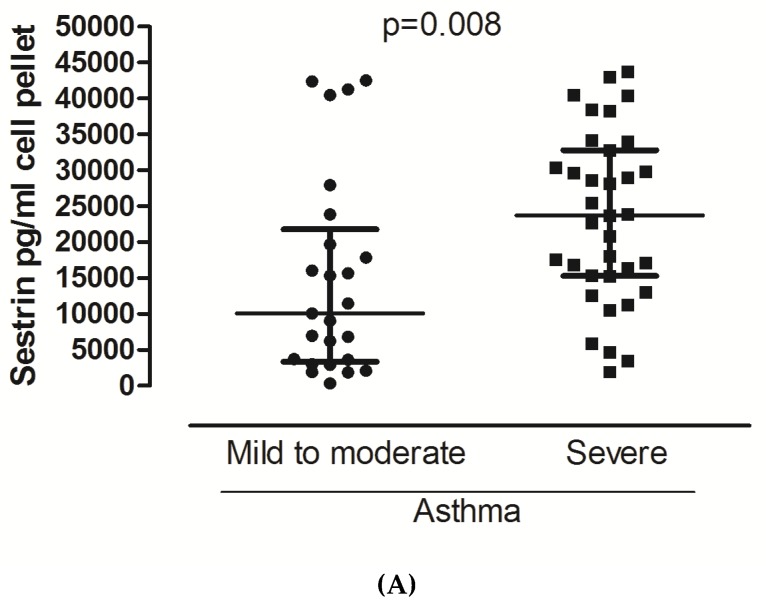
Sestrin 2 levels in patients with mild to moderate and severe asthma in cell pellets (**A**) and sputum supernatants (**B**). For data, see the table.

**Figure 2 jcm-09-00473-f002:**
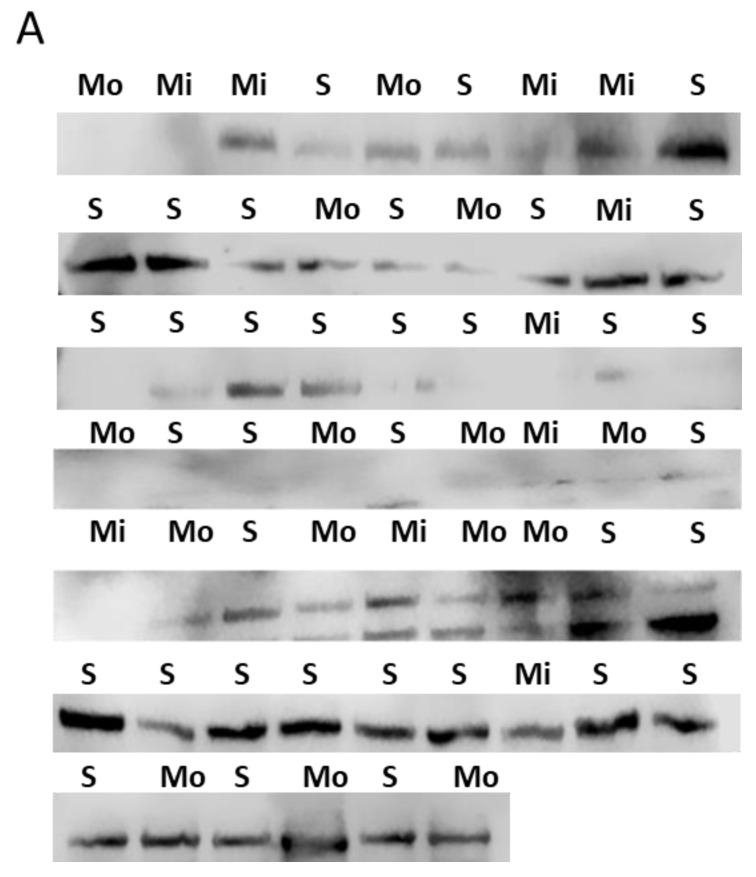
Serum SIRT1 levels in patients with mild to moderate and severe asthma. (**A**) Western blot images detecting SIRT1 in serum. Mi: mild asthma, Mo: moderate asthma, and S: severe asthma. (**B**) Individual plot of a SIRT1 band density (relative optical density) in Western blot. For data, see text.

**Table 1 jcm-09-00473-t001:** Subjects characteristics.

Variables	All *n* = 60	Mild to Moderate Asthma *n* = 25	Severe Asthma *n* = 35	*p*-Value*
Age	54 ± 13	56 ± 14	54 ± 13	0.876
Gender F/M	44/16	18/7	26/9	0.466
Atopy	24	11	13	0.186
Smoking	12	5	4	ND
BMI kg/m^2^	27 ± 5	26 ± 4	27 ± 6	0.435
FEV_1_ %	83 ± 23	92 ± 21	77 ± 22	0.011
FEV_1_/FVC	73 ± 13	75 ± 9	71 ± 15	0.136
Neutrophils %IS	16(3,29)	9(2,25)	19(3,38)	0.021
Eosinophils % IS	1(0,6)	1(0,2)	3(2,7)	0.031
Treatment	60	25	35	ND
ICS	35	0	35	
High doses ICS ^				
LABA	56	23	33	
LAMA	13	0	13	
LTRAs	17	7	10	
Omalizumab	5	0	5	
Mepolizumab	6	0	6	
Oral Cs ^^	8	0	8	

Data are presented as mean ± standard deviation (SD) or as median (interquartile ranges). Abbreviations: BMI = body mass index, FEV_1_ = forced expiratory volume in one second, IS = induced sputum, ICS = inhaled corticosteroids, LABA = long-acting β2 agonists, CS = corticosteroids, LAMA= long-acting muscarinic antagonists, and LTRA: leukotriene receptor antagonists. ^ >800 μg budesonide/day or equivalent. ^^ six patients were receiving 5 mg prednisolone/day, while two were receiving 7.5 mg prednisolone/day, and *statistically significant differences between mild to moderate vs severe asthma. *p* < 0.05 represents a statistically significant difference.

**Table 2 jcm-09-00473-t002:** Inflammatory variables in different biological samples.

Variables	Mild to Moderate Asthma	Severe Asthma	*p*-Value
**Endocan**			
Serum pg/mL	15 (11, 29)	21 (10, 34)	0.315
Sputum supernatant pg/mL	28 (25, 35)	29 (24, 34)	0.976
**Sestrin**			
Serum pg/mL	2645 (927, 3928)	1745 (1090, 3496)	0.519
Sputum supernatant pg/mL	7476 (4265, 9273)	9524 (5696, 12,373)	**0.029**
Cell pellet pg/mL	10,084 (3349, 21,784)	23,748 (15,280, 32,742)	**0.008**

Data are presented as median (interquartile ranges). Bold letters indicate statistically significant difference.

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
