# Peer review of "Sirtuin 1: Endocan and Sestrin 2 in Different Biological Samples in Patients with Asthma. Does Severity Make the Difference?"

_jcm, 2020, doi:10.3390/jcm9020473_

Round 1

Reviewer 1 Report

Main comments:

The materials and methods section is well detailed. Nevertheless, the authors should describe in more detail on what basis patients with severe asthma are distinguished from those with moderate asthma.

Western blot Image must be shown.

Are there difference in terms of inflammatory cells in the sputum of moderate compared to severe asthma patients?

Experiments to explain the role of Sestrin 2 in severe asthma must be conducted, to increase the scientific soundness of the manuscript.

Minor comments:

-Check the font size (line 47-58-60).

-Line 15: remove the bold to “background”

-In Table 1: What does ** means?

-Line 176: specify what “OCS” stay for

Author Response

1. The materials and methods section is well detailed. Nevertheless, the authors should describe in more detail on what basis patients with severe asthma are distinguished from those with moderate asthma.

Response: Patients with severe asthma are distinguished from those with moderate asthma on the basis of treatment with high doses of ICS. [page2, lines79-80].

2. Western blot Image must be shown.

Response: We provide the western blot image as figure 2A. Figure legend for Figure 2 was respectively changed.

3. Are there difference in terms of inflammatory cells in the sputum of moderate compared to severe asthma patients?

Response: In table 1 we clearly show the requested data. Statistical significance was also provided

4. Experiments to explain the role of Sestrin 2 in severe asthma must be conducted, to increase the scientific soundness of the manuscript.

Response: We agree with reviewers’ comment and we have already added a brief comment in the conclusion section [Page 9, lines 302-303].

 Minor comments:

-Check the font size (line 47-58-60).

Response: Font was checked.

-Line 15: remove the bold to “background”

Response: Bold was removed

-In Table 1: What does ** means?

Response: ** was removed.

-Line 176: specify what “OCS” stay for

Response: OCs was abbreviated as following: Oral corticosteroids.

Reviewer 2 Report

Dear Sir/Madam,

The manuscript is very well written. There may be some re-wording of the technical terms in order to allow the average reader to understand the topic. Other than that, the manuscript is publishable in the JCM Journal.

Author Response

We thank the reviewer for his/her positive approach

Round 2

Reviewer 1 Report

 Accept in present form

This manuscript is a resubmission of an earlier submission. The following is a list of the peer review reports and author responses from that submission.

Round 1

Reviewer 1 Report

Introduction section should be expanded, to better understand the aim of the study. The authors should better explain the rationale that prompted them to investigate the expression of these three targets.

The materials and methods section is well detailed. Nevertheless, the authors should describe in more detail on what basis patients with severe asthma are distinguished from those with moderate asthma.

Data described in the follow sentences, has to be shown: “No significant correlations were detected between serum SIRT1 levels and lung function nor between SIRT1 and the inflammatory cells in induced sputum.”

Are there difference in terms of inflammatory cells in the sputum of moderate compared to severe asthma patients?

“neither Endocan nor SIRT1 levels differed between patients with different asthma severity” this means that the authors detected these biomarkers, but they did not find differences in terms of biomarkers levels. But in the next sentence they affirm: “We failed to detect SIRT1 in both cell pellets and supernatants while Sestrin 2 and Endocan were detectable in all samples.” So, what is the truth?

The conclusion is not supported by the results. The authors show only an increased expression of Sestrin 2 in the sputum of patients with severe asthma. All that is associated with its role is speculative. On which basis the authors state that: “Sestrin 2 is increased in patients with severe asthma as part of a mechanism that may modify structural alterations through the imbalance between oxidative stress and antioxidant activity.

Experiment supporting this has to be conducted.

Reviewer 2 Report

The study is focused on evaluating the role of specific biomarkers in severity of asthma and whether there is a difference in their levels among patients with severe asthma compared to patients with mild to moderate asthma. The authors do a very well job in describing the background and the design of the study. Sestrin 2 is the only biomarker found to be different and higher in severe asthma. I believe the discussion is well written and includes enough reference to the results of other studies done on same biomarkers although somewhat limited and from animal studies. Wonder if there are more other human studies with similar design on these biomarkers that can be added to the discussion which I believe can improve the manuscript.